# Using diagnostic data from veterinary diagnostic laboratories to unravel macroepidemiological aspects of porcine circoviruses 2 and 3 in the United States from 2002–2023

Guilherme Cezar[1], Edison Magalhães[1‡], Kinath Rupasinghe[1], Srijita Chandra[1‡], Gustavo Silva[1], Marcelo Almeida[1‡], Bret Crim[1‡], Eric Burrough[1], Phillip Gauger[1‡], Christopher Siepker[1‡], Marta Mainenti[1‡], Michael Zeller[1‡], Eduardo Fano[1], Pablo Piñeyro[1], Rodger Main[1‡], Mary Thurn[2‡], Paulo Lages[2‡], Cesar Corzo[2‡], Albert Rovira[2‡], Hemant Naikare[2‡], Rob McGaughey[3‡], Franco Matias-Ferreyra[3‡], Jamie Retallick[3‡], Jordan Gebhardt[3‡], Jon Greseth[4‡], Darren Kersey[4‡], Travis Clement[4‡], Angela Pillatzki[4‡], Jane Christopher-Hennings[4‡], Melanie Prarat[5‡], Ashley Johnson[5‡], Dennis Summers[5‡], Craig Bowen[6‡], Joseph Boyle[6‡], Kenitra Hendrix[6‡], Andreia G. Arruda[7‡], Daniel Linhares[1], Giovani Trevisan[1]*

1 Veterinary Diagnostic and Production Animal Medicine, College of Veterinary Medicine, Iowa State University, Ames, Iowa, United States of America, 2 Veterinary Population Medicine, University of Minnesota, Saint Paul, Minnesota, United States of America, 3 College of Veterinary Medicine, Kansas State University, Manhattan, Kansas, United States of America, 4 Veterinary & Biomedical Sciences Department, South Dakota State University, Brookings, South Dakota, United States of America, 5 Ohio Animal Disease and Diagnostic Laboratory, Reynoldsburg, Ohio, United States of America, 6 College of Veterinary Medicine, Purdue University, West Lafayette, Indiana, United States of America, 7 Department of Veterinary Preventive Medicine, College of Veterinary Medicine, The Ohio State University, Columbus, Ohio, United States of America

☯ These authors contributed equally to this work.
‡ EM, SC, MA, BC, PG, CS, MM, MZ, RM, MT, PL, CC, AR, HN, RM, FMF, JR, JG, JG, DK, TC, AP, JCH, MP, AJ, DS, CB, JB, KH, and AGA also contributed equally to this work.
* trevisan@iastate.edu

## Abstract

Porcine circoviruses (PCVs), including porcine circovirus 2 (PCV2) and porcine circovirus 3 (PCV3), have been associated with clinical syndromes in swine, resulting in significant economic losses. To better understand the epidemiology and clinical relevance of PCV2 and PCV3, this study analyzed a dataset comprising diagnostic data from six veterinary diagnostic laboratories (VDLs) in the United States of America. The data comprised of polymerase chain reaction (PCR) test results, sample type, and age group for PCV2 and PCV3 submissions from 2002–2023. Findings indicated a decrease in the percentage of PCV2-positive submissions after introducing a commercial PCV2 vaccine in 2006 and a resurgence in positivity after 2018, particularly in breeding herds, associated with an increased number of submissions using processing fluid samples. After its first report in the U.S. in 2016, PCV3 detection had an upward trend in the percentage of positive cases, peaking in spring 2023. PCV3 detection was more frequent in adult/sow farms, while PCV2 was more frequently detected in the wean-to-market category. An additional analysis used results from tissue

**Data Availability Statement:** The SDRS project has legal confidentiality agreement with participant VDLs posing restrictions on sharing raw data publicly. The data used to generate information for this manuscript regarding PCV2, PCV3, and disease diagnosis is publicly available on the SDRS webpage (https://fieldepi.org/sdrs/). Additional data may be made available upon reasonable request and approval by SDRS participant institutions; please direct your request to SDRS email sdrs@iastate.edu and the corresponding author.

**Funding:** This project was supported by the Swine Health Information Center (SHIC) grants # 21-120, 22-002, and 23-062 (https://www.swinehealth.org/). This project was partly supported by the Agriculture and Food Research Initiative Competitive Grant no. 2023-67015-39883 from the USDA's National Institute of Food and Agriculture. The funders had no role in study design, data collection and analysis, decision to publish, or preparation of the manuscript.

**Competing interests:** The authors have declared that no competing interests exist.

diagnostic data from 2019–2023 from one VDL to associate PCR cycle threshold (Ct) values with the probability of confirming a PCV2 or PCV3 disease diagnosis confirmation. An interpretative PCR Ct cutoff for PCV2 and PCV3 diagnoses was assessed based on the logistic regression model associating Ct values with the presence of tissue lesions. The analysis considered only cases tested for PCV2 and PCV3 by PCR with tissue evaluations by diagnosticians. An interpretative Ct cutoff of 22.4 for PCV2 was associated with a high probability of confirming a diagnosis of PCV2 clinical disease through histopathology. For PCV3, the interpretative cutoff with the highest performance was 26.7. These findings contribute to the ongoing efforts to monitor and understand the clinical relevance of PCV2 and PCV3 PCR results, identifying potential disease challenges.

## Introduction

Porcine circoviruses (PCVs) belong to the *Circoviridae* family and were first identified as a contaminant of pig kidney cell lines in the 70s [1] and later in the 90s as the cause of a multisystemic wasting disease in weaning pigs in North America, Europe, and Asia [2]. Since then, PCVs, with the exception of PCV1, which has not been described as a pathogenic virus, have been associated with various syndromes in swine, including systemic, reproductive, enteric, and respiratory [3–5]. PCVs are ubiquitous in domestic swine and have been reported globally, and researchers have reported PCV1, PCV2, and PCV3 detections in the United States (U.S.) [2, 6, 7]. Porcine circovirus-associated diseases (PCVAD) [8, 9] can cost producers an average of $3–4 per pig in economic losses [9, 10], demonstrating the importance of monitoring and controlling these pathogens in swine farms.

Since the description of the first PCV2 clinical cases, serum samples have been used to monitor PCV2 viremia and antibody levels in swine [11]. Then, in 2008, oral fluid was identified as an effective population-based sample to monitor PCV2 presence, allowing for the detection of viral DNA through polymerase chain reaction (PCR) in samples collected from the growing pig population [12]. In the farrowing phase, processing fluids proved efficient for monitoring herds for the presence of porcine reproductive and respiratory virus (PRRSV) [13] and demonstrated their capability of detecting PCV DNA through PCR in the suckling piglet population [14].

However, a PCR-positive test alone is insufficient to confirm PCV2 clinical disease without additional diagnostic tests. A combination of factors, such as piglet immunity, farm management practices, and the route of infection, affect the expression of PCVAD [15]. Instead, a triad of factors, including decreased animal growth and histopathological lesions associated with the presence of the virus in primarily lymphoid tissues, is necessary to diagnose PCVAD [16]. Empirical observations suggest that the likelihood of clinical disease increases as the PCR Ct values decrease [17].

PCV3 has been detected in the U.S. since 2016 [7], including the co-detection with PCV2 in clinical samples [18]. The first PCV3 report in the U.S. associated the virus with reproductive failure, and porcine dermatitis and nephropathy syndrome (PNDS) in sows [7]. Also, clinical reports described an association between PCV3 and multisystemic inflammatory processes in perinatal and weaned pigs, with poor growth performance and mortality in the field [4, 19, 20]. In Brazil, lymphohistiocytic myocarditis, myositis, and gliosis in suckling pigs was a common finding in clinical cases where PCV3 was detected through in situ hybridization and PCR in litters born with caudally rotated ears [21]. Even though PCV3 has been associated with

several clinical presentations [22], demonstrating the clinical relevance of this virus remains a challenge.

Daily, various samples are submitted to veterinary diagnostic laboratories (VDLs) for PCR testing to identify the genetic material of numerous pathogens. Diagnostic test results remain stored in the VDLs' Laboratory Information Management Systems (LIMS). The Swine Disease Reporting System (SDRS, https://www.fieldepi.org/sdrs) project was established to collect, standardize, and aggregate anonymized diagnostic data results of the participant VDLs on a real-time basis, monitor those data, and report information on the activity of endemic porcine agents in the U.S. Using the PRRSV as a model, the SDRS created an extensive database that identifies emerging and re-emerging animal health threats that are rapidly reported to the swine industry [23, 24].

Currently, six veterinary diagnostic laboratories (VDLs) accredited by the American Association of Veterinary Laboratory Diagnosticians (AAVLD) and members of the National Animal Health Laboratory Network (NAHLN) participate in the SDRS project. One participant, VDL, also shares confirmed porcine tissue disease diagnosis, including PCVAD cases [25, 26].

Due to the complexity of monitoring PCV clinical cases, there is a need to develop tools to reveal and monitor changing patterns of PCV2 and PCV3 detection in swine farms. Using aggregated PCR cases reported by the VDLs that can detect PCV2 and PCV3 in real-time to unravel the megatrends of these viruses in the U.S. has never been conducted to the knowledge of the authors. This study aimed to evaluate the macroepidemiological aspects of aggregated PCV2 and PCV3 real-time PCR data over time and establish the real-time capability to rapidly identify changes in PCV2 and PV3 detection patterns. A secondary objective aimed to investigate the association between PCV-positive PCR cycle threshold (Ct) values and confirmed PCV disease diagnosis in submissions, using data from one participant VDL, including tissue evaluation by a diagnostician.

## Materials and methods

### Data collection and handling

This epidemiological study was performed with porcine diagnostic data gathered from six VDLs: Iowa State University (ISU), University of Minnesota, and Kansas State VDLs; South Dakota State University Animal Disease Research & Diagnostic Laboratory (ADRDL), Ohio Animal Disease Diagnostic Laboratory (ADDL), and Purdue University ADDL. Following the methodology described in previous studies [23, 27], anonymized clientele PCR submission data, test and test results were recovered from the six VDLs, cleaned, and collated into a standardized format at the submission level. All submissions had laboratory clientele information, including the owner's information, farm, veterinarian, and submitter, anonymized before sharing with the SDRS project. The original SDRS data processing used a SAS script (SAS® Version 9.4, SAS® Institute, Inc., Cary, NC) to collate the diagnostic data. Data processing and collation were modified to use a web-based application written in C# 10 (C#, Microsoft, Redmond, WA) leveraged by the NET 6 framework. Final processed data were stored in a secure Microsoft SQL server database hosted at the ISU Veterinary Diagnostic and Production Animal Medicine Department server. The final PCV2 and PCV3 PCR collated data were further plotted and analyzed to reveal both pathogens' detection trends over time by PCR testing results, specimen type, animal age category, PCR Ct values, and geographic location, i.e., state where the sample was collected.

### PCR data collation

Collated data were organized at a submission level using the VDL-assigned accession ID at the time of submission as a unique identifier. For the purpose of this study, a "case" is considered

as all the samples related to a unique accession ID. For time identification, the received date reported by the VDLs was utilized. The site state provided in the submission forms was used as a geographic location. Samples identified as originating from research facilities (farm type), vaccine samples or cell culture (specimen), non-porcine, and from other countries, not the United States (e.g., Mexico, Canada, etc.), were removed from the database.

The Logical Observation Identifiers Names and Codes (LOINC®, https://loinc.org/) and Systematized Nomenclature of Medicine Clinical Terms code (SNOMED CT, https://www.snomed.org/value-of-snomedct) were used to standardize analyte testing and test results, and specimen recording. Using LOINC® and SNOWMED CT improves the likelihood of data aggregation being accurately recorded and collated. Standardized nomenclature of the LOINC® and SNOWMED CT codes allows the aggregation of related tests or specimens irrespective of VDL-specific test names, allowing standardization of data exchange among laboratories. Submissions with more than one specimen tested had specimens labeled as "multiple".

PCV2 and PCV3 submissions from 2002 to 2023 were collated using distinct accession IDs. PCR results reported by the VDLs (positive, negative, suspect, or inconclusive) were used to establish the final case result in the database. For cases to be considered positive, at least one sample within the case was required to be PCR-positive. Alternatively, negative cases had to have all samples with negative PCR testing results. Suspect and inconclusive cases were reported according to each laboratory criteria.

Month and year were extracted from the sample received date. Next, a season was assigned using the month and year variables. The months June, July, and August had a season designated as "summer"; September, October, and November as "fall"; December, January, and February as "winter"; and March, April, and May as "spring." For this research, a full-year cycle corresponds to the 4-seasons that started on December 1st and ended on November 30th of the subsequent year.

A variable age category was created based on the farm type, age unit, and age variables provided by the VDL's submissions. If the farm type was provided, the age category was assigned based on this variable (e.g., suckling piglets, breeding herd, nursery, grow-finish, replacement, boar stud). When the farm type was not provided, the age category was established based on the age/age unit of the animals (0–21 days as suckling piglets; 22–63 as nursery; 64–200 as grow-finish; > 200 as adults). Then, the age categories were aggregated into phases: adult/sow farm (breeding herds, replacement, boar stud, suckling piglets, and adults) and wean-to-market (nursery and grow-finish). Submissions lacking information regarding farm type, age, or age unit were categorized with age category and phase assigned as "unknown".

The Ct values of the PCR results were aggregated using the positive samples within a laboratory submission. C# scripts calculate the average, minimum, and maximum PCR-positive Ct values aggregated by unique accession ID. For example, a submission with three positive samples for PCV2 with Ct values of 34, 33, and 23 had an average, minimum, and maximum Ct value assigned as 30, 23, and 34, respectively. The number of positive samples was assigned based on distinct sample IDs within a case. In cases where the sample ID had more than one result for the same analyte, such as retesting the same sample, the most recent reported result date was retained.

## Data analysis and visualization

The final collated dataset access was established with R algorithms (R, v.4.2.3, R Core Team) by connecting directly to the centralized database and managed on a local computer for data analysis. The dataset was then connected to Microsoft Power BI (Power Business Intelligence; Microsoft, Redmond, WA), enabling a user-friendly data visualization. Charts developed in

Power BI allowed further exploration of patterns in the dataset, using features such as filled maps, line charts, and clustered columns to reveal the megatrends of PCV2 and PCV3 PCR detection over time, state, specimen, and age category. For the specimen data visualization and analysis, all the tissue submissions that were not lung, lymph node, spleen, and fetus, or only had the description "tissue" with no specification were classified as "tissue others" This nomenclature was created to account for specimens such as liver, kidney, heart, pancreas, brain, and tonsils representing less than 1% of the tissue data and were assigned with the specimen described as "tissue others".

## PCV2 and PCV3 Ct value association with confirmed tissue disease diagnosis

Submissions from 2019–2023 with a PCV2, PCV3, or both confirmed tissue diagnosis from only one participant laboratory, i.e., the ISU-VDL, were retrieved and used to create a curve for the probability of having a confirmed disease diagnosis in tissue cases based on the Ct value obtained from PCR testing in tissue samples, i.e., excluding non-tissue samples such as oral fluid, processing fluids, serum that represents cases where the diagnostician did not have the opportunity to evaluate tissues for a complete diagnostic evaluation. A confirmed diagnosis at the ISU VDL was assigned by diagnosticians using a standardized diagnostic code (Dx code) nomenclature system [25]. The Dx code comprises four components and records the system (respiratory, digestive, systemic, etc.), insult (bacteria, parasite, virus, etc.), lesion (myocarditis, pneumonia, pleuritis, etc.), and etiology or disease [25]. A Dx code is assigned for submissions containing tissue and supported by diagnostic evidence associated with the case, e.g., clinical history, laboratory tests and test results performed for the case, and macroscopic and microscopic lesions. The relationship between the PCV2 or PCV3 Dx code and PCR Ct values was explored by comparing Ct value ranges and assigned PCV Dx codes.

Logistic regression models using the PCV2 or PCV3 Dx code (yes or no Dx code) as the explanatory variable and the PCR Ct value as the response variable calculated the probability of a submission having a PCV2 and PCV3 confirmed diagnosis for each decimal increment in the Ct range of PCVs tissue cases also tested by PCR. ROC curves were built for each model, calculating the area under the curve (AUC) to evaluate model performance. For the purpose of this study, the true positive (TP) cases were established for submissions with a Ct value under the defined cutoff with a PCV2 or PCV3 Dx code assigned, and the true negative (TN) cases were the cases above the cutoff with no Dx code assigned. False negatives (FN) were cases with a Dx code but Ct above the cutoff, and the false positives (FP) were the cases with Ct below the cutoff but with another etiology/disease Dx code assigned (Table 1).

The recall ($\frac{TP}{(TP+FN)}$) and accuracy ($\frac{(TP+TN)}{(TP+TN+FP+FN)}$) of the model were assessed for Ct values as a predictor of disease confirmation in submissions containing tissues. Based on this study, interpretative PCR cutoffs were established to suggest that once a PCR passes a Ct value, the result, while technically positive, is unlikely to be associated with disease (i.e., a disease-associated

**Table 1. Example of the 2x2 table for calculating the recall, accuracy, false positive, and false negative values.**
X = Ct value number.

|  | PCV Dx code assigned | |  |
| --- | --- | --- | --- |
| Ct value below (X) | Yes | No |  |
| Yes | TP | FP | TP+FP |
| No | FN | TN | FN+TN |
|  | TP+FN | FP+TN | Total |

cutoff). This interpretative PCR cutoff was calculated for PCV2 and PCV3 using the recall and accuracy of the logistic model from the total cases filtered from the Dx code database and tested for PCV2 PCR and PCV3 PCR. Retained submissions were further classified as having a PCV2, PCV3, or both Dx code assigned or etiology/disease Dx code assigned.

The interpretative PCR Ct cutoff was established based on the maximum recall and accuracy intersection, where both values reach the minimum difference. Each analysis was conducted separately for PCV2 and PCV3. For PCV2 cases, the urogenital system cases were filtered from the Dx code database separately from other systems (respiratory, systemic, cardiovascular, nervous, and digestive) to calculate a specific interpretative PCR Ct cutoff for abortion cases (fetuses). Compared with PCV2, PCV3 had a lower number of cases and analysis focusing on the urogenital system could not be performed for PCV3.

## Results

### PCV2 and PCV3 trends of detection by PCR

The final PCR database comprised 154,984 PCV2 cases from 2002 to 2023. The first PCV3 PCR data was recovered in 2016 and had a total of 49,975 cases tested up to December 2023. The generated and aggregated information was made publicly available in an online visualization platform at the SDRS project (SDRS, http://www.fieldepi.org/SDRS).

From 2002 until 2017, PCV2 had an average of 5,971 cases per year, with the highest number of cases submitted in 2015, at 7,888 (Fig 1). The average number of PCR cases increased by 91.52% from 2018 to 2023, with an average of 10,817 cases submitted yearly. An increase in the number of positive cases by PCR was also observed in the same time frame, with an increase of 74.50% (2,569 to 4,483) (Fig 1). Regarding the percentage of PCV2-positive PCR cases, the peak occurred in 2006 at 75.28% (4,395 of 5,838). The average percentage of positive cases from 2002 to 2006 was 59.49%, and from 2007 to 2011, a sharp decrease started to occur,

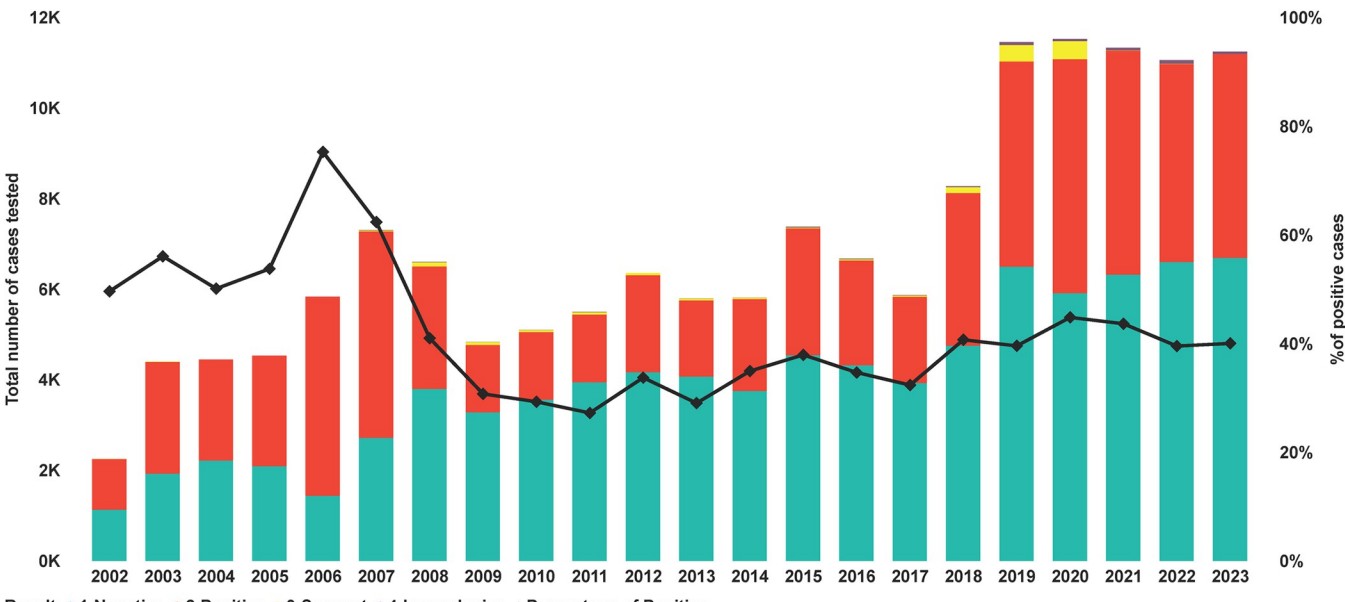

**Fig 1. Results of PCV2 cases tested by PCR over time (2002–2023).** Each point (x-axis) represents a year. The bars represent the number of cases tested and color-coded by testing results where red was positive, green negative, yellow suspect, and purple inconclusive (Y-axis). The black line represents the percentage of positive cases (secondary Y-axis).

decreasing from 75.28% (4,395 of 5,838) in 2006 to 27.24% (1,498 of 5,496) positive cases in 2011 (Fig 1). From 2012 to 2017, the average positive cases slightly increased to 34.04% (12,879 of 37,825), and 2015 had the largest percentage of PCV2 positive in this 6-year period with 38.12% of positive cases (2801 of 7,368). From 2018 to 2023, the average percentage of positive cases increased by 7.35% compared with 2012 to 2017, reaching 41.39% (26,898 of 64,891).

PCV3 was first detected in the U.S. in 2016, and VDLs started reporting test and test results for this pathogen in the fall of 2016. PCV3 testing became more frequent in the winter of 2018. From 2016–2018, the PCV3 PCR developments and validation were described, and commercial PCR kits became available [28, 29]. From the winter of 2018 until the winter of 2019, there was an increase in the number of cases by 365% (661 to 2,417) (Fig 2). Since then, the average number of PCV3 cases was 2,405 per season, with the highest number of cases (3,562) submitted and tested in spring 2023. In 2018, participant VDLs started offering multiplex PCR testing capable of detecting PCV2 and PCV3 within the same diagnostic assay. The average percentage of PCV3-positive cases from 2017 until spring 2018 never reached over 40% (Fig 2). However, after this period, PCV3 positive detection rate increased, having an average of 50.06% of cases per season, with peaks of increased positivity in the spring of 2020 with 61.54% (1,624 of 2639) and 2023 at 61.19% (1,670 of 2,726) of cases.

With the implementation of the PCV2 and PCV3 multiplex PCR in 2018–2020 by participant VDLs, PCV2 and PCV3 have been tested concomitantly with more frequency. In 2017, only 71 cases were tested for PCV2 and PCV3 concomitantly; in 2023, 8,817 cases were tested for both pathogens (Fig 3). This number represents 78.35% (8,817 from 11,252) of all PCV2 cases tested for PCR in 2023. Considering cases tested for PCV2 and PCV3 by PCR, the percentage of positive cases for both PCV2 and PCV3 also increased. In spring 2018, 14.81% (60 of 405) of the cases tested positive for both pathogens and in the spring of 2023, this number increased to 33.51% (909 of 2,713). Most of the positive cases for PCV2 and PCV3 from 2017 to 2023 were from processing fluids at 40.26% (3,924 of 8,180), oral fluids at 22.22% (1,818 of 8,180), and lung at 13.27% (1,086 of 8,180).

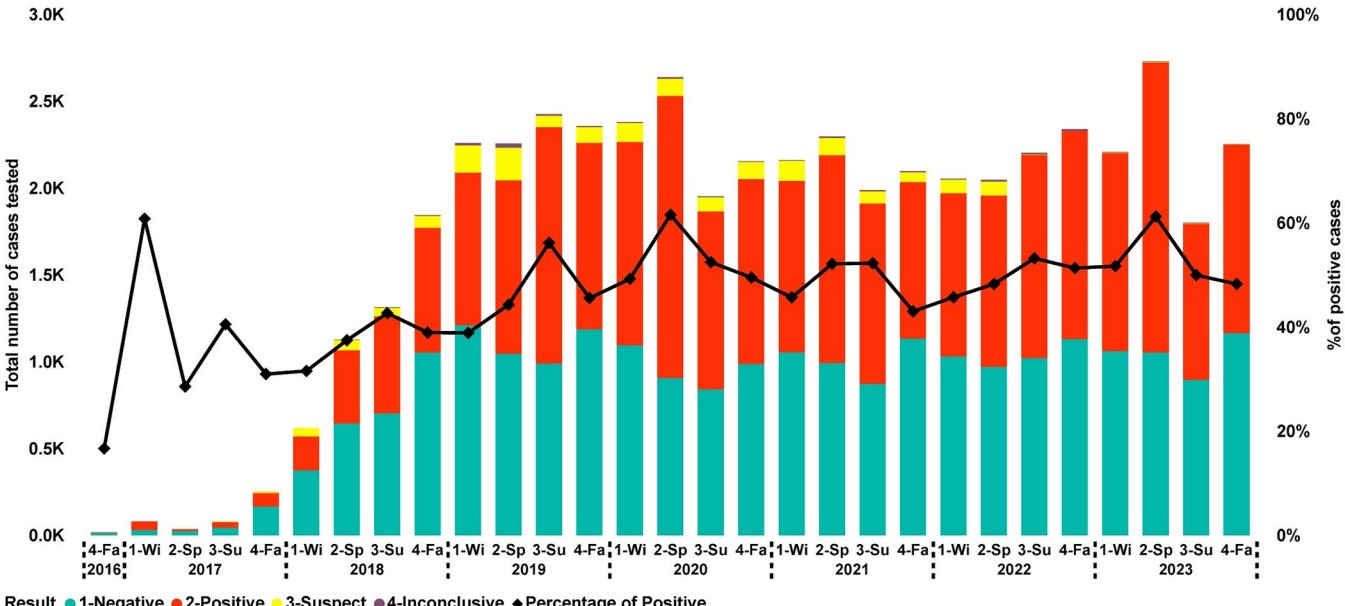

**Fig 2. Results of PCV3 cases tested by PCR over time (2017–2023).** Each point (x-axis) represents a season (1-Wi: Winter; 2-Sp: Spring; 3-Su: Summer; 4-Fa: Fall) within a year. The bars represent the number of cases tested and color-coded by testing results where red was positive, green negative, yellow suspect, and purple inconclusive (Y-axis). The black line represents the percentage of positive submissions (secondary Y-axis).

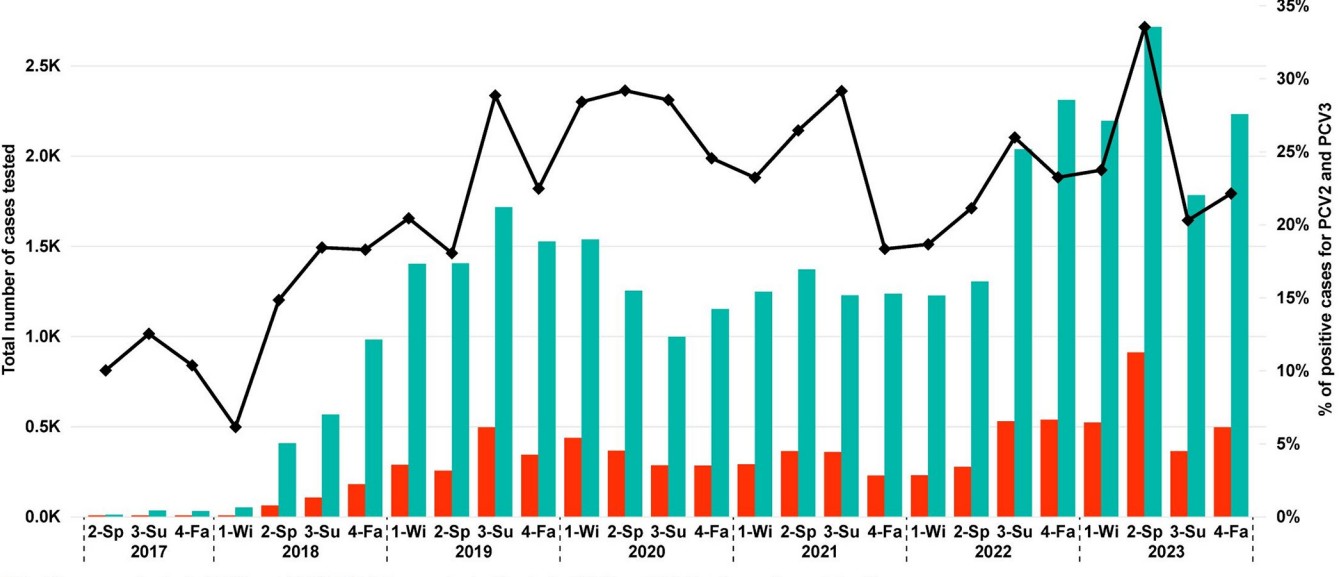

**Fig 3. Number of cases tested for both PCV2 and PCV3 and number of cases and percentage of positive cases for both analytes over time (2017–2023).** Each point (x-axis) represents a season (1-Wi: Winter; 2-Sp: Spring; 3-Su: Summer; 4-Fa: Fall) within a year. The green bars represent the total number of cases tested for PCV2 and PCV3, and the red bars represent the total positive cases for both PCV2 and PCV3 on the same case (Y-axis). The black line represents the percentage of positive submissions for both PCV2 and PCV3 on the same case (secondary Y-axis).

Wean-to-market was the pig production phase with the highest number of cases tested for PCV2: 39.71% (61,554 of 154,989), and adult/sow farm was the phase with the highest number of cases tested for PCV3: 41.33% (20,658 of 49,975). Since the summer of 2017, the PCV3 percentage of positive cases in adult/sow farms has consistently been above the percentage of positive cases in the wean-to-market phase (Fig 4A and 4B). For PCV2, the percentage of positive cases in the wean-to-market phase was predominantly above or equal to the PCV2 percentage of adult/sow farm positive cases, with the only exception being spring 2023 (Fig 4A and 4B). Regarding the percentage of positive cases, PCV3 had a higher percentage of positive cases in the adult/sow farm category every year compared to PCV2 (Fig 4A). On the wean-to-market cases, PCV2 had a higher percentage of positive cases than PCV3, except in winter and spring 2017 (Fig 4B). For both PCV2 and PCV3, no obvious visual seasonal patterns of detection were identified in this study.

For cases tested for PCV2, the age category "unknown" represented 34.40% (53,319 of 154,989) of cases, followed by grow-finish category 23.75% (36,796 of 154,989), and nursery 15.68% (24,313 of 154,989). For PCV3, the highest number of cases were in the age category grow-finish with 22.32% (11,158 out of 49,975), followed by breeding herd with 21.74% (10,869 out of 49,975) and suckling piglets with 18.58% (9,289 of 49,975). Improvements in the data capturing of farm type, age, and age unit information have been performed by VDLs since the percentage of "unknown" age categories decreased over time. From 2002–2010, the percentage of the "unknown" age category was 73.46%% (34,295 of 46,681) for PCV2 cases, and since then, this number has decreased to only 8% (871 of 10,877) of the cases categorized as unknown in 2023. In 2023, 26.24% (2,854 of 10,877) of the PCV2 cases were from grow-finish animals, representing most of the cases.

The specimen most submitted for PCV2 and PCV3 PCR testing was "tissue others" (All the tissue cases that were not lung, lymph node, spleen, and fetus). Within the "tissue (others)," 99.38% (66,813 of 67,226) were labeled as "tissue" or "tissue homogenate" without specifying

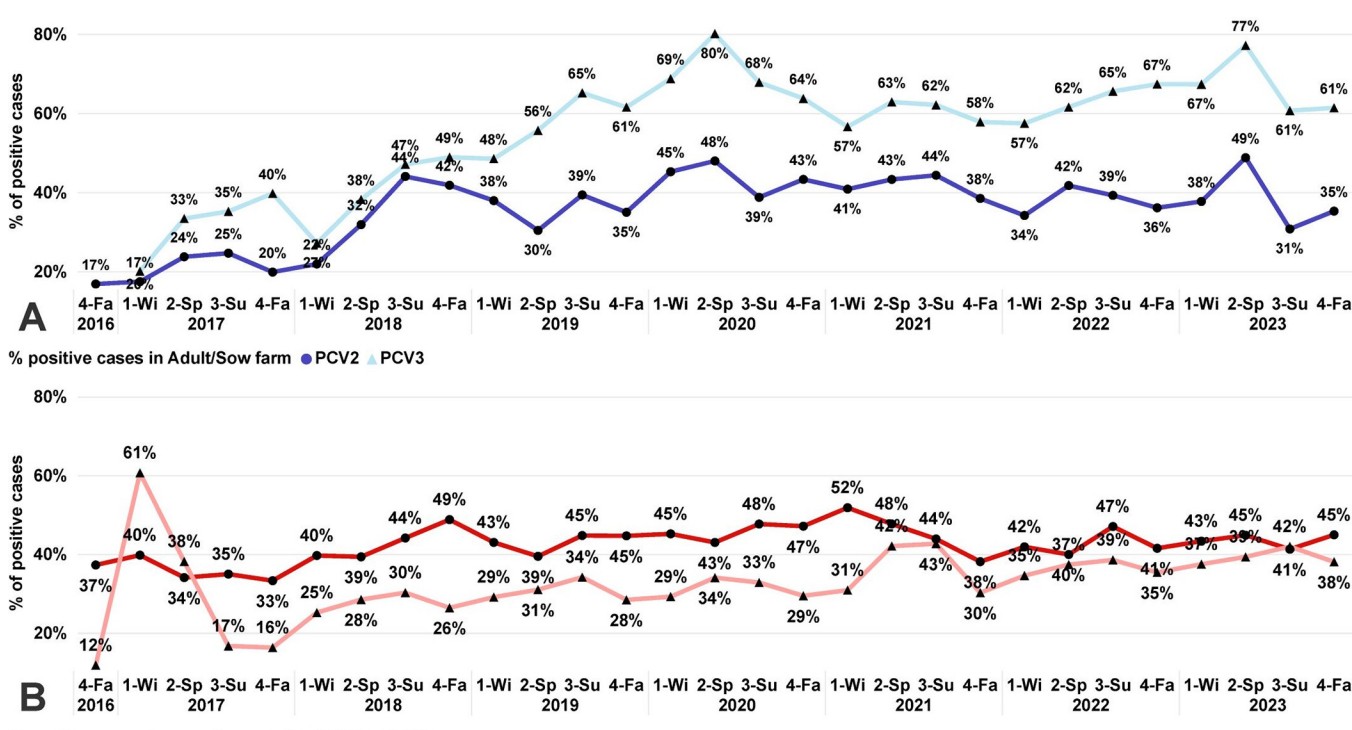

**Fig 4. Percentage of positive cases tested for PCV2 and PCV3 in adult/sow farm and wean-to-market categories over time (2017–2023).** Each point represents a season (1-Wi: Winter; 2-Sp: Spring; 3-Su: Summer; 4-Fa: Fall) within a year (x-axis). **A**: Percentage of positive cases in Adult/Sow farm. **B**: Percentage of positive cases in wean-to-market.

the tissue origin. Processing fluids and lungs were the two specimen most submitted for PCV2 and PCV3 besides tissue (others). Tissue (others) was 43.40% (67,266 of 154,989), lung 12.82% (19,871 of 154,989), and serum 11.96% (18,538 of 154,989) of the cases tested for PCV2 (Fig 5A). However, over time, the proportion of "tissue (others)" decreased sharply from 98.91% (2,276 of 2,301) in 2002 to 10,48% (1,191 of 11,355) in 2023. Additionally, the number of spleens tested for PCV2-PCR increased since 2015, when 0.15% (9 of 5,872) of the specimens tested were spleen, and in 2023, the number increased to 9.56% (1,076 of 11,355). Processing fluids were 27.39% (13,691 of 49,975), tissue (others) 19.87% (9,934 of 49,975), and lung 14.79% (7,392 of 49,975) of the cases tested for PCV3 (Fig 5B). In 2018, processing fluids samples started to be tested for PCV2 and PCV3 by PCR, and since 2019, they have become the specimen most submitted and tested for both circoviruses. On the other hand, the percentage of tissue samples started to decrease over the years since 2006, when 89.7% (5,476 of 6,125) were tissue samples, reaching the lowest percentage of 10.48% for PCV2 cases (1,191 of 11,355) and 13.26% for PCV3 cases (1,187 of 8,950) in 2023.

## Confirmed PCV2 and PCV3 tissue cases

From the ISU-VDL data including a disease diagnosis confirmation based on tissue evaluation, 14,915 cases were tested for PCV2, PCV3, or both and had a Dx code assigned and were retained for the PCR interpretative Ct cutoff analysis. There were 2,800 cases with PCV2 tissue confirmed disease diagnosis, 371 with PCV3, and 69 with both PCV2 and PCV3 confirmed diagnosis. Overall, 37.76% of the cases (5,629 of 14,915) were positive by PCV2 PCR testing, and 17.66% (2,634 of 14,915) were positive for PCV3 by PCR. From all accessions with a PCV3

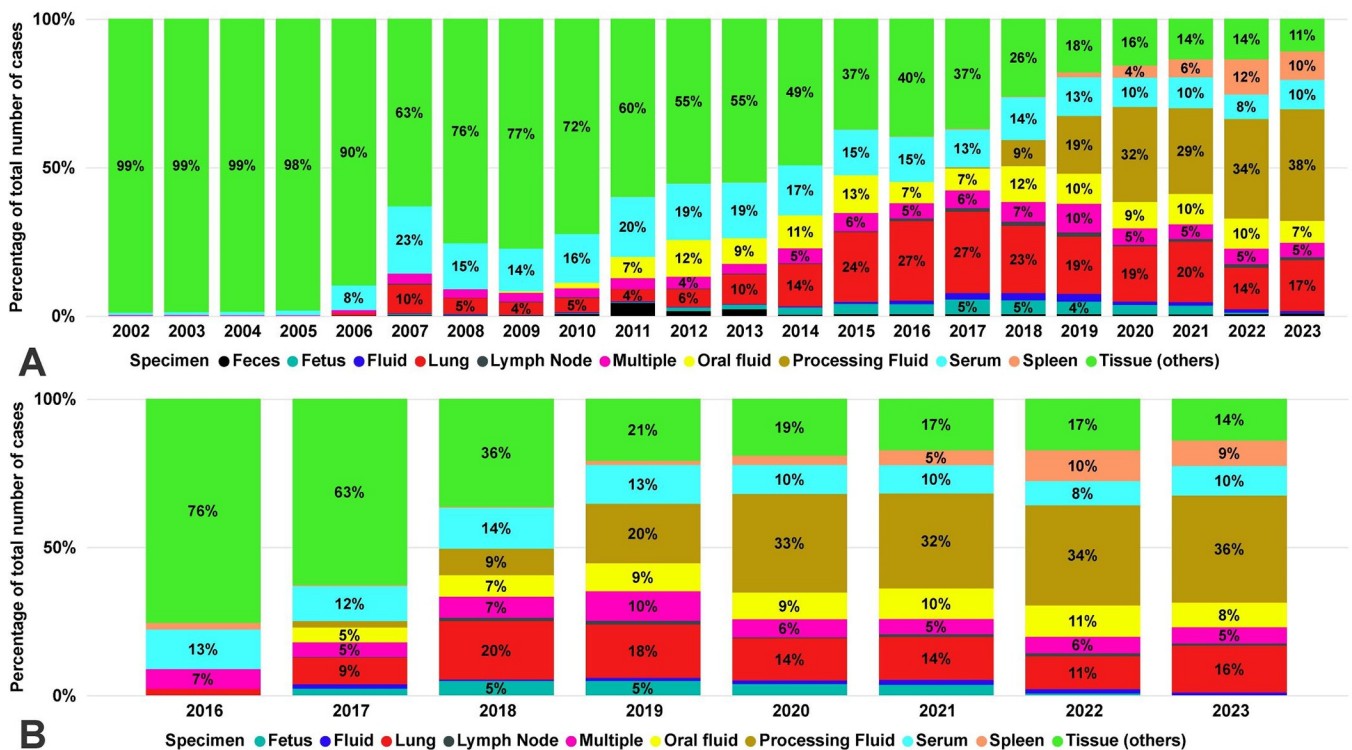

**Fig 5. Proportion of specimens tested for PCV2 and PCV3 by PCR over time. A:** Each bar represents a year, and each color is a specimen type tested for PCV2 PCR. **B:** Each bar represents a year, and each color is a specimen type tested for PCV3 PCR.

Dx code, 13.40% (59 of 440) were PCR-positive for both PCV2 and PCV3. Regarding all accessions with a PCV2 Dx code, 10.71% (300 of 2,800) were PCR-positive for both PCV2 and PCV3. A total of 2,421 cases had a PCV2 Dx code and were positive for PCV2 PCR. Among PCV2 cases positive for PCV2 PCR and had a PCV2 Dx code, the specimens most frequently submitted were lung at 56.67% (1,372 of 2,421), tissue (others) at 30.15% (730 of 2,421), and lymph nodes at 3.09% (75 of 2,421). The PCV3 PCR positive associated with a Dx code had 393 cases, and the most frequent specimens were tissue (others) at 34.86% (137 of 393), lung at 33.84% (133 of 393), and fetus at 14.75% (58 of 393).

Regarding the cases with Dx code and positive for PCV2 or PCV3 PCR, the range of Ct values for the cases analyzed was from 5 to 36.8. The AUC calculated from the ROC of the PCV2 and PCV3 logistic models using Dx codes were 0.89 and 0.77, respectively. The total cases filtered for calculation of accuracy and recall of the interpretative PCR cutoff analysis were 5,629 for PCV2 and 2,634 for PCV3. The recall and accuracy values for each PCR Ct unit from 10–35 are represented graphically in Fig 6. The optimal value is the intersection where the accuracy and recall lines meet in the plot, and these PCR Ct values were 22.4 for PCV2 (Fig 6A) and 26.7 for PCV3 (Fig 6B). PCV2 interpretative cutoff (Ct = 22.4) had an accuracy of 81.63% and a recall of 81.15%. For PCV3, the interpretative cutoff (Ct = 26.7) had an accuracy of 70.69% and a recall of 70.99%. Also, for PCV3 cases, the false positive rate of the interpretative PCR CT cutoff of 20 was 1.51%, representing that a few cases did not have PCV3 Dx codes when the Ct value was below 20 (34 of 2,634). Regarding the PCV2 urogenital cases filtered, the interpretative PCR Ct cutoff with higher performance was 24, which had a recall of 81.68% and an accuracy of 81.12%. These cases represented 12.91% (727 of 5,629) of total PCV2 cases, but 54.31% (233 of 429) of total breeding herd cases.

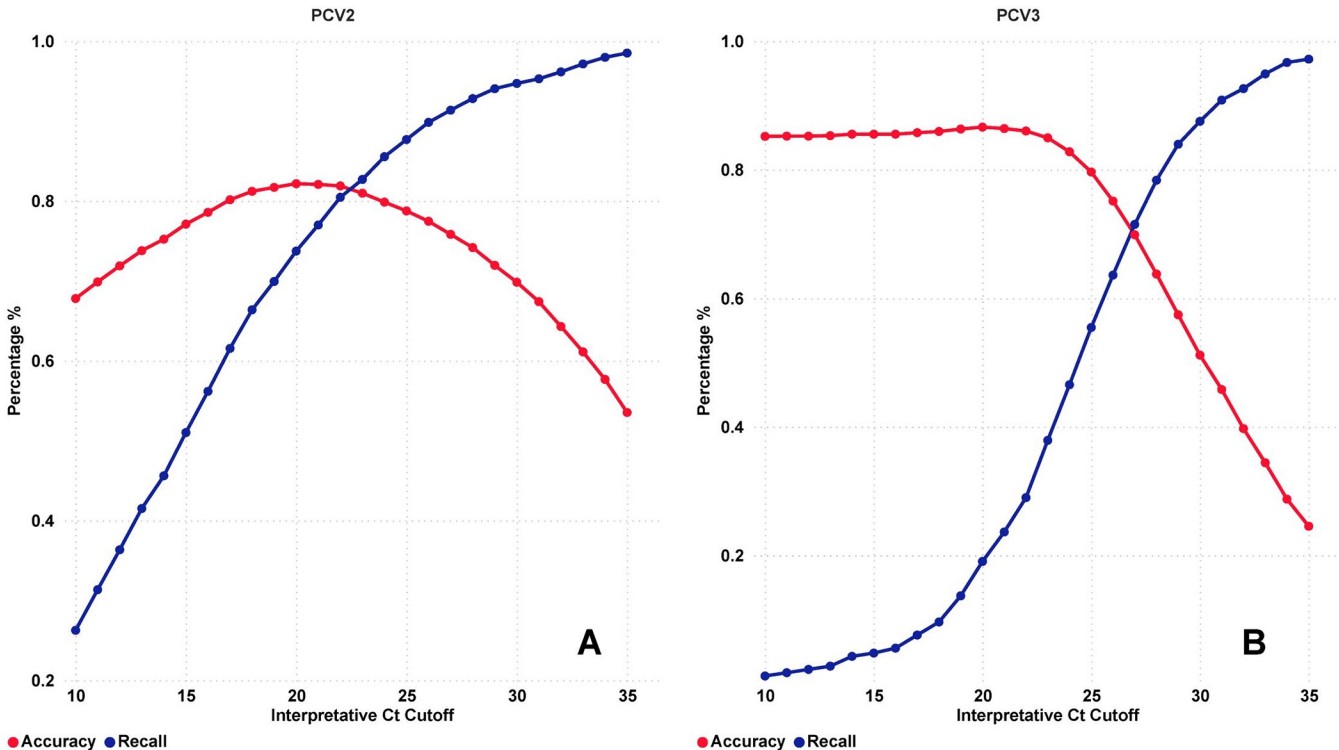

**Fig 6. Optimal Ct cutoff curve for PCV2 and PCV3 cases with diagnosis confirmed on tissue evaluation. A**: Accuracy and recall of each average PCR Ct value of PCV2 cases with/without diagnosis confirmed on tissue evaluation. **B**: Accuracy and recall of each average PCR Ct value of PCV3 cases with/without diagnosis confirmed on tissue evaluation.

## Discussion

This study revealed key macroepidemiological aspects of PCV2 and PCV3 detection by PCR using diagnostic data from six U.S. VDLs. The standardized data was successfully integrated and has continuously reported PCV2 and PCV3 PCR detection patterns data through the SDRS channels, informing stakeholders of current trends for these pathogens. Additionally, VDL data regarding PCV2 and PCV3 continues to be used proactively to monitor and inform significant changes in detection trends.

Our findings described an increased percentage of PCR-positive cases for PCV2 from 2002–2006, and a sharp decrease in the percentage of positive cases from 2007–2011. In 2006, PCV2-inactivated vaccines became commercially available and were rapidly accepted globally, resulting in decreased PCV2 morbidity, which may have resulted in the decreased PCV2 PCR positive cases due to vaccine implementation [30, 31]. Also, the reduction in clinical signs due to the vaccine implementation might have affected veterinary interest in PCV2 diagnosis in upcoming years. Additionally, from 2012–2018, there were changes in the prevalence of PCV2 genotypes in the U.S., with PCV2 genotype D becoming the most prevalent [30, 32]. This genotype drift might have affected the detection rate, reflected in a slight increase in the positive detection rate observed in our data during this period. In 2018, there was another increase in the PCV2 PCR positive rate, which could be associated with the adoption of processing fluids as a sample type to test suckling piglets for PCV2, demonstrating high sensitivity for detecting systemic viruses [13, 14]. Since then, processing fluids have become the most frequent specimen submitted for PCV2 PCR testing, representing 38% of all submitted specimens in 2023. Although PCV2 DNA can be detected by PCR in processing fluids, more research is

required to understand the clinical relevance of positive processing fluids and the impact on the sow herd, suckling piglets, or downstream growing pigs [14].

PCV3 had a higher percentage of positive cases derived from the phase adult/sow farms compared with wean-to-market, whereas PCV2 was the opposite with a higher percentage of positive cases in the wean-to-market phase compared with adult/sow farms. The higher detection of PCV2 in wean-to-market phase over adult/sow farm is supported by the common period of infection described in the literature, either for PCV2 subclinical and clinical cases, occurring in animals between 4–11 weeks of age [33]. Longitudinal studies also support a high percentage of positive animals in the grow-finish population, reporting over 73% of positive serum samples, with animals consistently having positive results from 5 to 22 weeks of age [34]. Additionally, our study demonstrated a higher PCV3 detection in the adult/sow farm category. Even though PCV3 is a virus associated with reproductive failures in breeding herds [7, 35], the difference in the percentage of positive cases was higher than 10% compared with PCV2 in each season since winter 2019. Further investigations are required to understand the dynamics of PCV3 in breeding herds to comprehend this increased PCR positive rates. In contrast to previous studies that identified a seasonal pattern in PRRSV [24] and enteric coronaviruses [24, 27], PCV2 and PCV3 data did not present seasonality.

A previous study reported 8.3% of PCV2/PCV3 co-detection at the case level during the period of 2016–2018 [18], slightly lower than the 12.03% reported in this study for 2016–2018. After 2018, the increased number of cases tested for both pathogens is reported in this study, which can be attributed to the implementation of a multiplex PCR for the simultaneous detection of both pathogens [36, 37] by the VDLs, which may have increased the frequency of testing of these pathogens within the same case. Another factor that contributes to co-detection is the use of processing fluids, which represented 40.26% (3,924 of 8,180) of the PCV2/PCV3 co-detection cases. PCV3 was first reported in the U.S. in 2016, and the increased number of cases tested for PCV3 in 2018 may be attributed to the availability of commercial PCR kits for PCV3 [28, 29]. The influence of the multiplex PCR on the number of cases is also supported by the increased proportion of PCV2 cases (33,839 of 64,907; 52.13%) tested for PCV3 in 2018–2023. Even though there is evidence of increased co-detection of PCV2 and PCV3, it is essential to recognize that merely identifying both analytes in the same case, especially where they are endemic, does not establish a causal link to the presence of disease associated with PCV2, PCV3, or both pathogens.

The specimens tested for both viruses were similar from 2018–2023, indicating that the criteria of clinical investigation for both PCV2 and PCV3 were similar. Data capture, standardization, and technology improvements in submission forms from the VDLs contributed to improved identifying of tissue specimens (e.g., lung, spleen, lymph node) more accurately in submissions from 2014 onwards [23, 26, 27]. However, within the "tissue (others)" specimen (i.e., other than lung, spleen, or lymph node), 99.38% (66,813 of 67,226) were labeled as "tissue" or "tissue homogenate" without specifying the tissue origin. This finding supports the necessity of improving specimen data capturing. From the population-based samples, oral fluids are recognized as more accessible to collect in farms, and with a high percentage of submissions submitted for other pathogens from the respiratory disease complex [23], oral fluids continued to be a specimen tested for PCV2 by PCR. The continuous usage of oral fluids might be attributed to its efficiency for monitoring swine pathogens, associated with the evidence of prolonged detection of PCV2 DNA using this sample type, and the correlation of positive results with possible clinical and subclinical PCVAD in the field [38, 39]. Even though the percentage of serum tested for PCV2 decreased over the years, it remained the fourth most tested specimen for PCV2 (1,079 of 11,355) and PCV3 (844 of 8,950) in 2023. Serum is still used for monitoring PCV2 and PCV3 dynamics in swine farms, for identification of viremic

animals, and evaluation of PCV2 vaccination programs using PCR and antibody assays [30, 33, 34].

Most of the confirmed disease diagnosis cases for both PCV2 and PCV3 had more PCR-positive results reported in lung tissues. Categorized as a pathogen belonging to the swine respiratory disease complex, PCV2 can be detected by PCR in pigs systemically affected by the virus and may cause pneumonia [40]. However, PCV3 researchers are still investigating the clinical impacts of this virus [41], and this higher detection on lung specimens raises the possibility of an association of PCV3 with respiratory clinical cases. It is essential to highlight that the histologic lesion was not necessarily identified on specimen positive in the PCR results, emphasizing the complexity of diagnosing these pathogens. Also, the low number of cases with co-diagnosis of PCV2 and PCV3 is similar to the frequency previously described in a retrospective study in the U.S. analyzing histologic lesions for both pathogens [42].

The association between the quantity of nucleic acid and PCVAD has been described for different clinical outcomes using serum [33, 43], fetuses [44, 45], and tissues [46]. Consistently with these findings, virus quantity higher than $10^7$ genomic copies per 500ng total DNA was associated with lesions or disease expression. The interpretative PCR Ct cutoff determined by the analysis of this study captured 81.53% (recall) of the overall cases with PCV2-confirmed tissue diagnosis that were below Ct 22.4 and 81.70% of the negative cases above Ct 22.4. Also, the interpretative Ct 22.4 had a false positive rate of 17.69% (1-Sp) and a false negative rate of 19.27% (1-Se). Based on the model's accuracy (81.63%), from 100 PCV2 positive PCR cases with an average Ct value below or above 22.4, less than 19 cases would have a misclassified PCV2 diagnosis. Also, in PCR-positive cases with Ct above 30, the likelihood of not receiving a Dx code (i.e., confirmed diagnosis on tissue evaluation) was 92.75%. These findings are supported by other studies suggesting that the viral load present in the sample may correlate with the expression of PCVAD in pigs [33, 43, 45]. For PCV2 reproductive cases, the interpretative Ct cutoff was 24, higher than the overall PCV2 tissue cases. The range of PCR cutoffs of reproductive cases in the literature varies among $10^7$ to $10^9$ genomic copies per 500ng total DNA [44–46]. However, it is essential to note that histologic confirmation of disease is particularly challenging in reproductive cases, partly due to significant post-mortem autolysis often present in fetuses, which may affect diagnostic interpretation in fetal tissues [45].

A pattern was identified for PCV3 cases with the PCR Ct value below 20. A low rate (1.15%) of false positive cases for PCV3 Dx code was identified below the Ct 20, suggesting that more investigation is necessary if the Ct is below this value. The optimal interpretative PCR Ct identified by the model for PCV3 was 26.7. The model recall detected 70.99% of cases with PCV3 Dx code. Also, a PCV3 PCR-positive case with a Ct value above 26.7 had a 93.28% probability of not receiving a confirmed PCV3 diagnosis on tissue evaluation. Even though the model had 70.69% accuracy, the false positive (29.36%) and negative (29%) rates for identified confirmed PCV3 diagnosis in tissue cases were high, which would create undesired alarms with no clinical cases below the Ct 26.7 and missing true clinical above this same PCR Ct value [47]. The high false positive rate and false negative rate in the tissue diagnosis are explained by the distribution of PCV3 cases with and without Dx code overlapping in terms of PCR Ct values, which makes it difficult for the model to establish an interpretative PCR Ct value with low false negatives and positives. Literature findings associating PCV3 clinical disease and PCR Ct values are complex, mainly due to the lack of a clear clinical presentation associated with its infection [22]. However, this study can support future research evaluating this association between clinical cases and PCR Ct values. Currently, a limitation of the PCV3 data is the low number of cases compared with PCV2 (440), which affects the final performance of the model.

A limitation of our study regards the usage of Ct values to determine cut-off. Ct values obtained from real-time PCR indirectly reflect the amount of virus present in a sample and

can vary significantly between PCR assays. Due to the nature of PCR amplification, these values often follow a logarithmic rather than a linear scale requiring normalization [48]. The Ct value interpretation is an inverse proportion scale with the viral load, whereas low the Ct values, the higher the viral load in the sample [49]. Consequently, caution should be exercised when averaging Ct values, as direct arithmetic means may not accurately represent the viral load [50]. In contrast, copy numbers are more easily standardized using a quantified reference sample and a standard curve. However, the genomic copies are obtained under different PCR testing assays, are not frequently requested by U.S. practitioners, and genomic copy data was not shared by participant laboratories. Such data can be a future accomplishment and development under laboratory results standardization and data sharing.

For the epidemiological interpretation of aggregated PCV2 and PCV3 tissue cases, the authors suggest that it is necessary to interpret the low average Ct values of tissue cases along with an increased number of cases and percentage of positive submissions. The association of these parameters coming from a robust dataset that covers > 95% of the swine samples submitted to the VDLs from the NAHLN can represent evidence of disease activity occurring in the field [24, 51, 52].

When aggregating veterinary diagnostic data from laboratories, it is important to acknowledge test results pertain to samples submitted for diagnostic testing and do not accurately reflect the occurrence, prevalence, or incidence of diseases in a particular region. This limitation arises from the passive nature of data collection at VDLs. Concluding disease incidence or prevalence from laboratory data is not advisable because the populations from which samples are drawn lack well-defined parameters, and the sampling process is generally not conducted randomly [27]. Determining disease status typically falls within the purview of attending veterinarians, who rely on a combination of medical history, clinical signs, animal characteristics, epidemiological data, laboratory findings, and pathological examinations to define the causal effect of a disease. Using aggregated PCR data can indicate abnormalities in pathogen detection in the field. Still, standard procedures will vary somewhat among laboratories in routine PCR testing, leading to potential discordant test results. This variability can be attributed to differences among technicians, extraction and/or amplification protocols, reagents, equipment, and other factors [53, 54] that can pose a limitation in interpreting aggregated PCR results from multiple VDLs. However, initiatives such as this project contribute to better standardization and further improvements in the laboratory network. Megatrends of detections revealed under this work, can be used to further guide research questions and design experimental or field-based trials to explore and clarify root causes of secular trends of changes in PCV2 or PCV3 detection.

This study unraveled the macroepidemiological aspects of PCV2 and PCV3 in the U.S. swine population. Additionally, a PCV2 monitoring tool based on the interpretative Ct cutoff results described in this research was implemented in the Swine Disease Reporting System project. Monitoring average Ct values of PCV2 tissue cases in the U.S. to alert when the average Ct of submissions is below 22.4 can aid producers and veterinarians in identifying when there is a higher probability of PCV2 activity in the field. The study also sheds light on PCV3 detection trends, contributing to further investigations regarding the virus dynamics.

## Acknowledgments

The authors of this manuscript acknowledge the VDLs clients for submitting samples for diagnostic testing. Also, we acknowledge past and current SDRS advisory group members for their insights regarding the field perspective of the analysis and volunteered time: Drs. Mark Schwartz, Paul Yeske, Deborah Murray, Brigitte Mason, Peter Schneider, Sam Copeland, Luc

Dufresne, Daniel Boykin, Corrine Fruge, William Hollis, Rebecca Robbins, Thomas Petznick, Kurt Kuecker, Lauren Glowzenski and Megan Niederwerder.

## Author Contributions

**Conceptualization:** Edison Magalhães, Gustavo Silva, Marcelo Almeida, Eric Burrough, Phillip Gauger, Christopher Siepker, Marta Mainenti, Michael Zeller, Eduardo Fano, Pablo Piñeyro, Rodger Main, Cesar Corzo, Albert Rovira, Hemant Naikare, Franco Matias-Ferreyra, Jamie Retallick, Jordan Gebhardt, Travis Clement, Angela Pillatzki, Jane Christopher-Hennings, Kenitra Hendrix, Andreia G. Arruda, Giovani Trevisan.

**Data curation:** Srijita Chandra, Bret Crim, Rodger Main, Mary Thurn, Paulo Lages, Rob McGaughey, Jon Greseth, Darren Kersey, Ashley Johnson, Joseph Boyle.

**Formal analysis:** Guilherme Cezar, Michael Zeller, Giovani Trevisan.

**Funding acquisition:** Guilherme Cezar, Rodger Main, Albert Rovira, Franco Matias-Ferreyra, Angela Pillatzki, Melanie Prarat, Craig Bowen, Kenitra Hendrix, Daniel Linhares, Giovani Trevisan.

**Investigation:** Eric Burrough, Phillip Gauger, Christopher Siepker, Marta Mainenti, Michael Zeller, Pablo Piñeyro, Hemant Naikare, Jordan Gebhardt, Jane Christopher-Hennings, Dennis Summers, Daniel Linhares, Giovani Trevisan.

**Methodology:** Guilherme Cezar, Kinath Rupasinghe, Gustavo Silva, Marcelo Almeida, Eric Burrough, Phillip Gauger, Christopher Siepker, Marta Mainenti, Michael Zeller, Eduardo Fano, Pablo Piñeyro, Cesar Corzo, Andreia G. Arruda, Daniel Linhares, Giovani Trevisan.

**Project administration:** Daniel Linhares, Giovani Trevisan.

**Resources:** Rodger Main, Albert Rovira, Hemant Naikare, Franco Matias-Ferreyra, Jamie Retallick, Travis Clement, Angela Pillatzki, Melanie Prarat, Craig Bowen, Kenitra Hendrix, Daniel Linhares, Giovani Trevisan.

**Software:** Kinath Rupasinghe, Srijita Chandra, Bret Crim, Mary Thurn, Paulo Lages, Rob McGaughey, Jon Greseth, Darren Kersey, Joseph Boyle.

**Supervision:** Eduardo Fano, Pablo Piñeyro, Daniel Linhares, Giovani Trevisan.

**Validation:** Pablo Piñeyro, Travis Clement, Dennis Summers, Craig Bowen, Andreia G. Arruda.

**Visualization:** Eduardo Fano, Jordan Gebhardt, Travis Clement, Ashley Johnson, Andreia G. Arruda, Giovani Trevisan.

**Writing – original draft:** Guilherme Cezar.

**Writing – review & editing:** Guilherme Cezar, Edison Magalhães, Gustavo Silva, Marcelo Almeida, Eric Burrough, Phillip Gauger, Christopher Siepker, Marta Mainenti, Michael Zeller, Eduardo Fano, Pablo Piñeyro, Cesar Corzo, Albert Rovira, Franco Matias-Ferreyra, Jamie Retallick, Jordan Gebhardt, Jane Christopher-Hennings, Melanie Prarat, Andreia G. Arruda, Daniel Linhares, Giovani Trevisan.

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
