## [Editor Report · Decision Letter 0]

24 Mar 2024

PONE-D-24-10038Using diagnostic data from veterinary diagnostic laboratories to unravel macroepidemiological aspects of porcine circoviruses 2 and 3 in the United States from 2002-2023.PLOS ONE

Dear Dr. Trevisan,

Thank you for submitting your manuscript to PLOS ONE. After careful consideration, we feel that it has merit but does not fully meet PLOS ONE’s publication criteria as it currently stands. Therefore, we invite you to submit a revised version of the manuscript that addresses the points raised during the review process.

We look forward to receiving your revised manuscript.

Kind regards,

Gianmarco Ferrara, PhD, MVD

Academic Editor

PLOS ONE

Journal Requirements:

   "This project was supported by the Swine Health Information Center (SHIC) grants # 21-120, 22-002, and 23-062 (https://www.swinehealth.org/). This project was partly supported by the Agriculture and Food Research Initiative Competitive Grant no. 2023-67015-39883 from the USDA’s National Institute of Food and Agriculture." 

5. Please remove your figures from within your manuscript file, leaving only the individual TIFF/EPS image files, uploaded separately. These will be automatically included in the reviewers’ PDF.

Additional Editor Comments:

The manuscript has two critical points reported by the system that should be addressed before the system goes for review. The first point concerns the title of the manuscript, which is very similar (although concerning a different pathogen) to the title of a recent manuscript by the same author. The second point concerns the number of authors. It is unthinkable that almost 40 authors have worked on a manuscript for which we invite the authors to include only and exclusively those who have made a significant contribution to the manuscript.

---

## [Author Response · Author response to Decision Letter 0]

1 Apr 2024

Dear Dr Gianmarco Ferrara, Academic Editor PLOS ONE

Thanks for your feedback on our manuscript “Using diagnostic data from veterinary diagnostic laboratories to unravel macroepidemiological aspects of porcine circoviruses 2 and 3 in the United States from 2002-2023”. Please find below the author's considerations for the items pointed out by the editors to consider our research for publication in the PLOS ONE journal: 

We verified the documents shared with us, and the authors of this manuscript believe it is in the right format for submission to PLOS ONE.

2) Please note that funding information should not appear in any section or other areas of your manuscript. We will only publish funding information present in the Funding Statement section of the online submission form. Please remove any funding-related text from the manuscript.

We have removed the funding information from the manuscript. 

3) Thank you for stating the following financial disclosure: 

"This project was supported by the Swine Health Information Center (SHIC) grants # 21-120, 22-002, and 23-062 (https://www.swinehealth.org/). This project was partly supported by the Agriculture and Food Research Initiative Competitive Grant no. 2023-67015-39883 from the USDA’s National Institute of Food and Agriculture." 

The statement is correct regarding the role of the funders in the study design. Therefore, we have included the sentence in the cover letter as requested.

4) We note that you have indicated that there are restrictions to data sharing for this study. PLOS only allows data to be available upon request if there are legal or ethical restrictions on sharing data publicly. For more information on unacceptable data access restrictions, please see http://journals.plos.org/plosone/s/data-availability#loc-unacceptable-data-access-restrictions. 

The Swine Disease Reporting System (SDRS) project, where the data used to produce this research work came from, has a legal confidentiality agreement with the participating veterinary diagnostic labs, restricting the sharing of raw data by the SDRS project. However, data sharing with additional institutions requires additional legal approval from the SDRS participant institutions. Therefore, we have edited the following language for the data availability section. “The SDRS project has legal confidentiality agreement with participant VDLs posing restrictions on sharing raw data publicly. The data used to generate information for this manuscript regarding PCV2, PCV3, and disease diagnosis is publicly available on the SDRS webpage (https://www.fieldepi.orf/SDRS). Additional data may be made available upon reasonable request and approval by SDRS participant institutions; please direct your request to SDRS email sdrs@iastate.edu and the corresponding author.” 

5) Please remove your figures from within your manuscript file, leaving only the individual TIFF/EPS image files, uploaded separately. These will be automatically included in the reviewers’ PDF.

We have removed all the images from the manuscript and already submitted all the figures in separate files.

6) The manuscript has two critical points reported by the system that should be addressed before the system goes for review. The first point concerns the title of the manuscript, which is very similar (although concerning a different pathogen) to the title of a recent manuscript by the same author. The second point concerns the number of authors. It is unthinkable that almost 40 authors have worked on a manuscript for which we invite the authors to include only and exclusively those who have made a significant contribution to the manuscript.

We kindly appreciate the sharing of such concerns by the PLOS ONE system. We respectfully submit our considerations. We truly believed that what PLOS ONE has identified as concerns were the primary reasons why we were targeting our cutting-edge article for consideration in PLOS ONE. Please see our specific: 

The article with a similar title, “Macroepidemiological aspects of porcine reproductive and respiratory syndrome virus detection by major United States veterinary diagnostic laboratories over time, age group, and specimen,” addressed a completely different pathogen, i.e., PRRSV. The methodology used for pCV2 and PCV3 has evolved from the PRRSV work and so uses similar terminology, e.g., the same terminology as “macroepidemiology” because it is the term that better defines the megatrends of detection analyzed in the manuscript. The manuscript for PRRSV was published as part of Dr. Trevisan's PhD and laid the foundation of a very successful project, i.e., the Swine Disease Reporting System, that has been kept alive for six years already. Dr. Trevisan is currently a faculty and mentors Dr. Cezar on his MS and has been able to expand the scope of SDRS to include PCV2 and PCV3 in the current manuscript, going beyond detection by PCR by also including interpretative diagnostic results of disease diagnosis in tissue cases for both endemic disease, i.e., PCV2; and a newly emerging disease, i.e., PCV3. At the time of PCV3’s emergence in the US, PLOS ONE was publishing our PRRSV manuscript. The PRRSV manuscript had significance and was for a period of time “among the top 10% most cited PLOS ONE papers published in 2019”. The authors believed that having a follow-up manuscript expanding the scope of the SDRS project on PLOS ONE would make a nice story and, therefore, have submitted it for consideration in PLOS ONE.

We agreed that the number of authors may be intimidating. In 2019, when we published our PRRSV article, we had 21 authors. At that time, four veterinary diagnostic laboratories were part of the project. Now, in 2024, the project has seven institutions that actively collaborate, six distinct veterinary diagnostic laboratories, and another university. Additionally, for the PCV2 and PCV3, there was a need to bring on board additional expertise on tissue disease diagnostics from one participant veterinary diagnostic laboratory. This project was only possible due to the extensive collaboration on data sharing, data analysis, manuscript writing and review, editing, and evaluation of methodology feasibility by the experts listed. Since it is a collaborative project, this research could not have been done without the participation of all the members. Again, we were very excited to tell a story of something that has grown over time in the same journal that initially accepted to publish our work. Therefore, we kindly ask you to reconsider your concern, and the property allows us to list and give credit as authors to those who contributed to this multidisciplinary research team. After providing evidence of high collaborative work that has evolved and provided science-driven advancements over time is “unthinkable”, as you mentioned in the editorial letter, and considered of no interest to PLOS ONE, please let us know, and we will reconsider our submission to this journal. 

Thank you for reviewing our manuscript and considering it for review. We appreciate your time and look forward to your response.

Regards,

Giovani Trevisan, DVM, MBA, PhD

Researcher Assistant Professor

Veterinary Diagnostic and Production Animal Medicine

Iowa State University College of Veterinary Medicine

Office: +1 515 294 3556

---

## [Decision Letter · Decision Letter 1]

24 May 2024

PONE-D-24-10038R1Using diagnostic data from veterinary diagnostic laboratories to unravel macroepidemiological aspects of porcine circoviruses 2 and 3 in the United States from 2002-2023.PLOS ONE

Dear Dr. Trevisan,

Thank you for submitting your manuscript to PLOS ONE. After careful consideration, we feel that it has merit but does not fully meet PLOS ONE’s publication criteria as it currently stands. Therefore, we invite you to submit a revised version of the manuscript that addresses the points raised during the review process.

We look forward to receiving your revised manuscript.

Kind regards,

Gianmarco Ferrara, PhD, MVD

Academic Editor

PLOS ONE

Reviewers' comments:

Reviewer's Responses to Questions

**Comments to the Author**

1. If the authors have adequately addressed your comments raised in a previous round of review and you feel that this manuscript is now acceptable for publication, you may indicate that here to bypass the “Comments to the Author” section, enter your conflict of interest statement in the “Confidential to Editor” section, and submit your "Accept" recommendation.

Reviewer #1: (No Response)

Reviewer #2: (No Response)

2. Is the manuscript technically sound, and do the data support the conclusions?

Reviewer #1: Partly

Reviewer #2: Yes

3. Has the statistical analysis been performed appropriately and rigorously? 

Reviewer #1: I Don't Know

Reviewer #2: Yes

4. Have the authors made all data underlying the findings in their manuscript fully available?

Reviewer #1: No

Reviewer #2: No

5. Is the manuscript presented in an intelligible fashion and written in standard English?

Reviewer #1: Yes

Reviewer #2: Yes

6. Review Comments to the Author

**Reviewer #1:** Guillerme et al.'s study primarily provides a descriptive analysis of the epidemiology of PCV2 and PCV3 in the U.S. swine population over time. Overall, it sounds like the paper has the potential to make a significant contribution to the field.

To improve the paper, it would be advantageous to incorporate more comprehensive details regarding the diagnostic tests employed and the interpretation of the results. This would enhance the clarity and depth of the study's findings. Additionally, it's important for the paper to clearly outline its limitations. This could include discussing potential biases introduced by the use of data from multiple VDLs, variations in laboratory protocols, and any other factors that may have influenced the results or interpretation of the findings. Providing a comprehensive overview of the study's limitations will help readers better understand the scope and implications of the research.

Please incorporate details regarding the diagnostic testing process. Given the existence of various protocols, it's unclear whether all samples underwent testing via real-time PCR TaqMan-based, Sybrgreen, or conventional PCR methodologies, especially during the initial years.

Have the VDLs ever conducted quantitative real-time PCR using the PCV2/PCV3 protocols? It would provide insight into the virus's quantity, as indicated by the Ct values. This information could prove immensely valuable and should be incorporated.

Incorporating PCV2 subtyping data would greatly enhance the value of the paper if it's accessible.

Were submissions from other countries excluded?

Please incorporate the study's limitations concerning variations in protocols across different laboratories, challenges in interpreting Ct values, and any other issues identified during the analysis.

Please ensure all references align accurately with the corresponding text.

The figures appear blurry.

Figure 1 employs the same color code for both negative and non-informed data points."

In lines 58-60, it would be more precise to specify that PCV1 is a non-pathogenic virus."

Lines 64-66: The term Porcine circovirus-associated diseases (PCVAD) typically refer to PCV2. The references should be reviewed accordingly. Additionally, the references regarding economic losses require correction.

Please ensure all references align accurately with the corresponding text.

In lines 109-110, it should be revised to 'PCV2 and PCV3 real-time PCR cycle threshold (Ct).

Regarding line 192, include more detail in the samples not included.

For line 188, please provide more details about the confirmed tissue disease. Was it confirmed via ISH or IHC?

171-173. CT values obtained from real-time PCR assays indirectly reflect the amount of virus present in a sample. Due to the nature of PCR amplification, these values often follow a logarithmic rather than a linear scale. Consequently, caution should be exercised when averaging CT values, as direct arithmetic means may not accurately represent the results.

**Reviewer #2:** The manuscript is well written and clear, the introduction adequate, methods and results well described and commented, and the amount of data used is impressive. However, I have some major concerns about the effect that secular trends in terms of veterinary perception, knowledge, diagnostic test availability, and changes in used matrices might have. These factors could be even more relevant than the underlying biological phenomena, biasing the results too much and preventing any useful conclusions.

Additionally, why was a Ct cut-off used instead of copy number? Ct values are highly variable among laboratories and assays, while copy numbers can be more easily standardized using a quantified reference sample and a standard curve. Was the reproducibility among laboratories assessed? Even if the Ct values were comparable among the laboratories involved in the study, the provided cut-off is not useful for other laboratories. On the contrary, providing such data could be misleading and lead to misdiagnoses.

More detailed comments and suggestions are reported in the attached PDF

7. PLOS authors have the option to publish the peer review history of their article (what does this mean?). If published, this will include your full peer review and any attached files.

Reviewer #1: **Yes: **Victor Neira

Reviewer #2: No

---

## [Author Response · Author response to Decision Letter 1]

6 Jun 2024

Dear PLOS ONE reviewers, 

Thanks for your feedback on our manuscript “Using diagnostic data from veterinary diagnostic laboratories to unravel macroepidemiological aspects of porcine circoviruses 2 and 3 in the United States from 2002-2023”. Please find below the author's considerations for the items pointed out by you to consider our research for publication in the PLOS ONE journal: 

Reviewer 1

1) To improve the paper, it would be advantageous to incorporate more comprehensive details regarding the diagnostic tests employed and the interpretation of the results. This would enhance the clarity and depth of the study's findings. Additionally, it's important for the paper to clearly outline its limitations. This could include discussing potential biases introduced by the use of data from multiple VDLs, variations in laboratory protocols, and any other factors that may have influenced the results or interpretation of the findings. Providing a comprehensive overview of the study's limitations will help readers better understand the scope and implications of the research.

We addressed the limitations of Ct values (lines 515-526 revised manuscript) and the different test procedures performed in veterinary diagnostic laboratories (lines 542-550 revised manuscript). Unfortunately, we cannot describe all the testing protocols in the manuscript, this information is not shared throughout the network built under the SDRS. Even though this is the case the participant laboratories follow standard procedures for accreditation through the American Association of Veterinary Laboratory Diagnosticians and the National Animal Health Laboratory Network (NAHLN). 

2) Please incorporate details regarding the diagnostic testing process. Given the existence of various protocols, it's unclear whether all samples underwent testing via real-time PCR TaqMan-based, Sybrgreen, or conventional PCR methodologies, especially during the initial years.

Unfortunately, we cannot describe all the testing protocols in the manuscript. This information is not shared throughout the network built by the SDRS; we understand that there is a limitation on comparing different diagnostic assays, but this is not under the scope of this work. Here, we look for the usage of reported test diagnostic results to identify what tested for and what was the PCR results for epidemiological work. However, we tried to address this limitation in the paragraph of lines 542-550 (revised manuscript) 

3) Incorporating PCV2 subtyping data would greatly enhance the value of the paper if it's accessible.

Yes, we agree with this addition. However, we are still developing the message system to receive the PCV2 ORF2 sequence data, which would allow the generation of information with genotype characterization. This might be a future analysis and development with genotype information and further sequence evaluation to identify genetic changes in these viruses over time.

4) Were submissions from other countries excluded?

Yes, submissions from other countries were excluded and we added that in the revised manuscript lines 142-144.

5) Please remove your figures from within your manuscript file, leaving only the individual TIFF/EPS image files, uploaded separately. These will be automatically included in the reviewers’ PDF.

We have removed all the images from the manuscript and already submitted all the figures in separate files.

6) Please incorporate the study's limitations concerning variations in protocols across different laboratories, challenges in interpreting Ct values, and any other issues identified during the analysis

As mentioned before, we incorporated the limitations in discussing the revised manuscript. Please evaluate lines 515-526.

7) Please ensure all references align accurately with the corresponding text.

References were realigned accurately. We double-checked the incorrect references and fixed them in the revised manuscript.

8) Figure 1 employs the same color code for negative and non-informed data points."

The NA data was an error in the coding process that was fixed and appropriately renamed as “Inconclusive.” A new image was updated within this resubmission.

 9) Lines 64-66: The term Porcine circovirus-associated diseases (PCVAD) typically refer to PCV2. The references should be reviewed accordingly. Additionally, the references regarding economic losses require correction.

The references regarding economic losses were fixed. Even though the terminology PCVAD is commonly used for PCV2, research papers were using the terminology also for PCV3 (Pathogenicity and immune modulation of porcine circovirus 3, Chen D et al. 2023). We added this reference next to the sentence, facilitating the readers' understanding.

10) Regarding line 192, include more detail in the samples not included.

Complementary information was added in the line 200-202. Please let us know if the information helped in the understanding.

11) For line 188, please provide more details about the confirmed tissue disease. Was it confirmed via ISH or IHC?

Not necessarily, the Dx code data has a limitation of not providing access to all the tests performed or requested by the diagnostician. A disease is coded as diagnosed if components in the diagnostic process involving submission story, macroscopic, microscopic, test and testing results provided evidence to support the diagnosis of a specific etiology/disease. In general, it follows the guidelines provided in Derscheid, et al, 2021, cited in our manuscript. Consequently, based on the accumulated evidence, the cases were analyzed by a trained diagnostician who assigned an etiology Dx code, such as PCV2 or PCV3. They might rely on the clinical history of the farm, clinical presentation of the case, and PCR results demonstrating a high number of PCV2/PCV3 genomic copies, IHC, ISH, and other tests performed. We understand the importance of addressing these individualities. However, we relied on the expertise of trained diagnosticians to look at each case individually and assign the diagnostic codes. Here, we provided the epidemiological context of how it compares when aggregating this massive amount of PCR and disease diagnosis data.

12) 171-173. CT values obtained from real-time PCR assays indirectly reflect the amount of virus present in a sample. Due to the nature of PCR amplification, these values often follow a logarithmic rather than a linear scale. Consequently, caution should be exercised when averaging CT values, as direct arithmetic means may not accurately represent the results.

As mentioned before, we incorporated the limitations in discussing the revised manuscript. Please evaluate lines 515-526, explaining the limitation of interpreting Ct values we added to the manuscript.

Reviewer 2

We went thorugh the PDF file and performed the changes addressed in the manuscript. Below there are the major comments made the reviewer and the answers from the presenting authors.

1) The manuscript is well written and clear, the introduction adequate, methods and results well described and commented, and the amount of data used is impressive. However, I have some major concerns about the effect that secular trends in terms of veterinary perception, knowledge, diagnostic test availability, and changes in used matrices might have. These factors could be even more relevant than the underlying biological phenomena, biasing the results too much and preventing any useful conclusions.

Additionally, why was a Ct cut-off used instead of copy number? Ct values are highly variable among laboratories and assays, while copy numbers can be more easily standardized using a quantified reference sample and a standard curve. Was the reproducibility among laboratories assessed? Even if the Ct values were comparable among the laboratories involved in the study, the provided cut-off is not useful for other laboratories. On the contrary, providing such data could be misleading and lead to misdiagnoses.

The authors tried to address the limitation of using diagnostic data, the possible biases on line 540, and the limitations of using Ct values on lines 515-526 of the manuscript with the importance for further developments to add genomic copies in the analysis. The number of genomic copies is not shared by participant laboratories and is performed under different diagnostic testing assays offered as a fee for service.

2) What were the case definition guidelines used to perform PCV2 and PCV3 associate disease diagnosis?

There are no specific guidelines for performing a PCV2 or PCV3 diagnosis; diagnostic characterization follows a myriad of factors, including submission story, macroscopic, microscopic, test, and testing results provided to support the diagnosis of a specific etiology/disease. Specifics of this process were described elsewhere and are not part of this manuscript (Derscheid, 2021). Therefore, cases were analyzed by a trained diagnostician, and based on all the evidence they had, they assigned an etiology Dx code as PCV2 and PCV3. If the diagnostician were unsure about the diagnostic, they would have assigned a Dx code as non-specified. 

3) what was the denominator? total samples or cases suspected of PCV2 infection? this might be of interest since it could reflect the ability of veterinarians in differential diagnosis

The denominator was described In lines 245-247 of the revised manuscript; the average number of tests increased from 5,971 (period of 2002 to 2017) to 10,817 (the years 2018-2023), which can impact the number of positive cases since more cases were being submitted. We reference figure 1 so the reader can follow the trend of the increased number of cases and the percentage of positive submissions.

4) See previous question, especially considering the change in diagnostic approach that probably led to test for PCV3 more samples than would have been if a specific PCR had to be used

Yes, we agree with this statement. In our discussion, we added the possible impact of the new development of duplex PCR (PCV2/PCV3) may have impacted the count and positivity of this virus on lines 446-448 on the revised manuscript. We can’t forget that PCV3 was just recently described, and it may have been present in the field for a longer time, it just has not been incorporated into the diagnostic routine before.

5) I believe that this is not the correct caption, or for some reason I can not visualized some of the lines mentioned in the caption. likely figure 3 and 4 have been inverted

Yes, it was inverted. We corrected the captions in the revised manuscript.

6) I would comment if the percentage of positive was on PCV2 PCR or total submitted samples. vaccine implementation and reduced clinical signs might have affected the veterinary perception and interest on PCV2 diagnosis

We added a sentence to the revised manuscript to better explain the discussion of the vaccine implementation and how much this can affect the veterinarian's testing strategies. Please check lines 408-410 of the revised manuscript.

7) Why was a Ct cut-off used instead of copy number? Ct values are highly variable among laboratories and assays, while copy numbers can be more easily standardized using a quantified reference sample and a standard curve. Was the reproducibility among laboratories assessed? Even if the Ct values were comparable among the laboratories involved in the study, the provided cut-off is not useful for other laboratories. On the contrary, providing such data could be misleading and lead to misdiagnoses.

Yes, we agree that the number of genomic copies would be more appropriate. Still, unfortunately, nowadays, the message system established by the VDLs does not include the information on genomic copies and not all of the PCR testing results are reported with genomic copies in the U.S. This is a limitation of our work that we addressed in the revised manuscript in the paragraph starting in line 515.

---

## [Decision Letter · Decision Letter 2]

26 Sep 2024

Using diagnostic data from veterinary diagnostic laboratories to unravel macroepidemiological aspects of porcine circoviruses 2 and 3 in the United States from 2002-2023.

PONE-D-24-10038R2

Dear Dr. Trevisan,

We’re pleased to inform you that your manuscript has been judged scientifically suitable for publication and will be formally accepted for publication once it meets all outstanding technical requirements.

Kind regards,

Gianmarco Ferrara, PhD, MVD

Academic Editor

PLOS ONE

Additional Editor Comments (optional):

All comments have been addressed

Reviewers' comments:

Reviewer's Responses to Questions

**Comments to the Author**

1. If the authors have adequately addressed your comments raised in a previous round of review and you feel that this manuscript is now acceptable for publication, you may indicate that here to bypass the “Comments to the Author” section, enter your conflict of interest statement in the “Confidential to Editor” section, and submit your "Accept" recommendation.

Reviewer #3: All comments have been addressed

2. Is the manuscript technically sound, and do the data support the conclusions?

Reviewer #3: Yes

3. Has the statistical analysis been performed appropriately and rigorously? 

Reviewer #3: Yes

4. Have the authors made all data underlying the findings in their manuscript fully available?

Reviewer #3: Yes

5. Is the manuscript presented in an intelligible fashion and written in standard English?

Reviewer #3: Yes

6. Review Comments to the Author

Reviewer #3: (No Response)

7. PLOS authors have the option to publish the peer review history of their article (what does this mean?). If published, this will include your full peer review and any attached files.

Reviewer #3: No

---

## [Editor Report · Acceptance letter]

29 Sep 2024

PONE-D-24-10038R2 

PLOS ONE

Dear Dr. Trevisan, 

I'm pleased to inform you that your manuscript has been deemed suitable for publication in PLOS ONE. Congratulations! Your manuscript is now being handed over to our production team.

Kind regards, 

on behalf of

Dr. Gianmarco Ferrara 

Academic Editor

PLOS ONE